# HDL in Immune-Inflammatory Responses: Implications beyond Cardiovascular Diseases

**DOI:** 10.3390/cells10051061

**Published:** 2021-04-29

**Authors:** Fabrizia Bonacina, Angela Pirillo, Alberico L. Catapano, Giuseppe D. Norata

**Affiliations:** 1Department of Pharmacological and Biomolecular Sciences, Università degli Studi di Milano, 20133 Milan, Italy; fabrizia.bonacina@unimi.it; 2Center for the Study of Atherosclerosis, E. Bassini Hospital, Cinisello Balsamo, 20092 Milan, Italy; angela.pirillo@guest.unimi.it; 3IRCCS MultiMedica, Sesto S. Giovanni, 20099 Milan, Italy

**Keywords:** high density lipoprotein, autoimmune disease, immune-inflammatory response, cholesterol efflux

## Abstract

High density lipoproteins (HDL) are heterogeneous particles composed by a vast array of proteins and lipids, mostly recognized for their cardiovascular (CV) protective effects. However, evidences from basic to clinical research have contributed to depict a role of HDL in the modulation of immune-inflammatory response thus paving the road to investigate their involvement in other diseases beyond those related to the CV system. HDL-C levels and HDL composition are indeed altered in patients with autoimmune diseases and usually associated to disease severity. At molecular levels, HDL have been shown to modulate the anti-inflammatory potential of endothelial cells and, by controlling the amount of cellular cholesterol, to interfere with the signaling through plasma membrane lipid rafts in immune cells. These findings, coupled to observations acquired from subjects carrying mutations in genes related to HDL system, have helped to elucidate the contribution of HDL beyond cholesterol efflux thus posing HDL-based therapies as a compelling interventional approach to limit the inflammatory burden of immune-inflammatory diseases.

## 1. Introduction

High-density lipoproteins (HDL) are commonly recognized as the “good carriers” of cholesterol in blood as they participate to cholesterol removal from the periphery (including vessel wall) to the liver for the excretion, in contrast to low-density lipoproteins (LDL), or “bad” cholesterol, that distribute dietary cholesterol from the liver to the periphery. The observation that HDL have been preserved during evolution and are the dominant lipoproteins in species that do not develop atherosclerosis and cardiovascular disease supports their role in modulating immunometabolic responses occurring at the vascular level and beyond [1]. The aim of this review is to provide a timely update on the involvement of HDL in the immune response, to discuss the molecular mechanisms that could explain this function beyond that of cholesterol cargo and to provide a rationale for targeting the HDL system in the setting of immune-inflammatory diseases.

## 2. HDL in Immune-Inflammatory Diseases: What Are HDL-C Levels Telling Us?

Plasma cholesterol levels are a robust and economic marker to predict cardiovascular risk in the population. Although many observational studies have demonstrated an inverse correlation between HDL-C levels and the risk of coronary heart disease and mortality, this association applies for values up to 80–90 mg/dL [2,3], whereas at higher levels a U-shaped curve is observed, with extremely high HDL-C levels being associated with an increased CVD risk [4]. In parallel, clinical studies with different pharmacological approaches, including niacin, fibrates, and cholesteryl ester transfer protein inhibitors, showed that a massive increase in plasma HDL-C levels does not translate in increased cardiovascular protection.

These findings lead to reconsider HDL-C not only as a CVD risk factor but rather as an indicator of other pathological conditions. Indeed, a U-shaped association has been reported not only for the incidence of CV events, but also for the incidence of infectious and autoimmune diseases [5,6]. On one hand, low HDL-C levels predict the risk of infection and autoimmune diseases, on the other hand, individuals with extremely high HDL-C levels are at increased risk of infectious diseases. Data from the ILLUMINATE trial showed that raising HDL-C with torcetrapib, a cholesteryl ester transfer protein (CETP) inhibitor, resulted in an increased risk of infectious disease-related deaths compared to placebo [7]. These findings highlight the connection between HDL-C levels/function, immune-inflammatory and infectious diseases. 

Patients with systemic lupus erythematosus (SLE) and rheumatoid arthritis (RA) are both characterized by an increased CVD risk due to the coexistence of dyslipidaemia and chronic inflammation. SLE patients present a classical “lupus lipoprotein pattern” with increased levels of VLDL and LDL and decreased HDL, a phenotype associated with accelerated atherosclerosis and cardiovascular death [8]. Reduced levels of HDL-C are also a common, despite not unequivocal, sign of other autoimmune diseases, including RA, skin diseases, such as psoriasis [9], and bowel diseases, such as Chron’s disease [10], especially during the active phase of the disease. While these are mainly descriptive studies, where HDL-C concentrations are used as a proxy of HDL levels, emerging approaches are trying to investigate other parameters that could better mark the number of HDL particles, including NMR analysis [11,12] or the measurement of the key structural protein of HDL, namely, apoA-I, whose levels are also decreased in patients with SLE [13].

### HDL Composition in Auto-Immune Diseases 

It has been widely proven that, in the presence of chronic cardiometabolic diseases or even under an acute pathological condition (such as infection or myocardial infarction), HDL may switch from a protective anti-inflammatory particle to a dysfunctional pro-inflammatory player [8,14,15,16,17,18,19,20,21]. HDL from patients with coronary heart disease exhibit a lower cholesterol efflux capacity (the leading anti-atherosclerotic function of HDL), a reduced endothelial-protective activity, paralleled by the acquisition of pro-atherogenic functions [22]. In type 2 diabetic patients, HDL show several structural alterations that include the enrichment with serum amyloid A (SAA), due to underlying low-grade chronic inflammation, the reduction of apoA-I content, and the increase in oxidized lipids and advanced glycation products [22]. Chronic kidney disease negatively impacts HDL structure and function, with the incorporation of the acute phase protein symmetric dimethylarginine into the HDL particle having a major negative consequence on HDL-mediated reverse cholesterol transport and endothelial protection [22].

A closer look at patients with autoimmune diseases evidence several modifications in the composition and functionality of HDL supporting a switch from an anti-inflammatory to a pro-inflammatory particle (piHDL) (Table 1) [8]. The major alteration is the displacement of apoA-I by SAA, together with the reduction of paraoxonase (PON1) activity; these changes affect the anti-inflammatory and anti-atherogenic properties of HDL [8]. Parallel to these changes in the proteome, substantial modifications of the HDL lipidome have been reported in SLE and RA patients. These include not only lower cholesteryl esters and more triglycerides [23,24], but also a reduced phospholipid content. The latter was reported in psoriatic patients but was also shown in other pathological conditions, including acute myocardial infarction, and is consistent with an elevated activity of secretory phospholipase A2 (sPLA2), as frequently observed under these circumstances [25]. In agreement with this, HDL from RA patients show an increased content of phospholipid species carrying arachidonic acid (C20:4 n-6), paralleled by the reduction of phospholipid species carrying omega-3 fatty acids (and more specifically docosahexaenoic acid, C22:6 n-3) [26]. Changes in the lipidome impacts also HDL function; indeed, cholesterol efflux capacity (CEC) [27,28] is reduced in SLE and RA patients [29] and is partially restored by treatment with tocilizumab, an interleukin-6 receptor inhibitor [30]. These findings further strengthen the relationship between systemic to cellular cholesterol handling and immune-inflammatory response [30].

## 3. The Role of HDL on Immune Cell Function

HDL possess several anti-atherogenic properties, ranging from anti-oxidant to the control of vascular tone, that are protective in the context of cardiovascular diseases [83,84]; however, the most important activity, which is also associated to several immune-modulatory effects, is represented by the ability of HDL to remove free cholesterol from cell membrane [85,86]. This pathway, that is either mediated by the interaction with specific cholesterol transporters, named ATP-binding cassette transporters ABCA1 and ABCG1, or can be transporter-independent, has been shown to shape the function of immune cells [87].

### 3.1. Anti-Oxidative, Anti-Inflammatory, and Vascular-Protective Effects of HDL

HDL are a cargo of proteins, enzymes, and molecules (from hormones to vitamins) that are not directly related to the maintenance of cholesterol homeostasis. Paraoxonase enzymes (PON1-3), for instance, confer anti-oxidative functions to HDL [88]. PiHDL from SLE patients show a reduction in the content of paraoxonase, resulting in increased oxidation of HDL and of their main apoprotein, apoA-I [89]. In addition, PON1 is also involved in the regulation of vascular tone by promoting endothelial nitric oxide synthase (eNOS) activation [90]; together with the inhibition of endothelial expression of adhesion molecules, reduction of apoptosis and the promotion of endothelial cell repair by hematopoietic precursors [91], PON1 contributes to the vaso-protective activity of HDL. On the contrary, oxidation of HDL could directly have pro-inflammatory effects on monocytes by promoting the upregulation of platelet-derived growth factor receptor β (PDGFRβ), increasing their chemotaxis and TNF-α release [45], but also enriching HDL with the pro-inflammatory SAA [45], as observed in SLE patients. Indeed, HDL from SLE patients failed to promote the synthesis and translocation of the nuclear factor ATF3, resulting in NF-κB activation and inflammatory cytokine production in macrophages [46]; this effect is mediated by the ability of oxidized HDL to bind lectin-like oxidized LDL receptor 1 (LOX1). It is worth to note, lipoprotein oxidation, included that of HDL, derives also from hyperactivated neutrophils of autoimmune patients: the release of active oxidative enzymes (including MPO, nitric oxide synthase and NOX) from their extracellular traps (NETs) results in the modification of HDL, that lose their anti-inflammatory and vasoprotective effects, as well as the ability to mediate reverse cholesterol transport [41]. Several other alterations in HDL function were associated with their oxidation [88,92,93,94].

Finally, other anti-inflammatory properties of HDL are associated to their ability to bind and neutralize the Gram-negative bacterial LPS [95] and Gram-positive bacterial lipoteichoic acid (LTA) [96], thus preventing generalized inflammatory responses. While LTA binds directly to HDL, LPS interacts with LPS-binding protein (LBP), an acute phase protein that shares similarities with CETP [97] and promotes LPS transfer from cell membranes to HDL, leading to LPS neutralization, probably via increased liver catabolism/elimination [98,99]. HDL participate to inflammatory response also by promoting the production of the acute phase protein PTX3 by endothelial cells [100]; PTX3, in turn, plays a double-edged sword role not only in infections [101], but also in atherosclerosis [102], thrombosis [103,104], and obesity [105]. Together these observations highlight the role of HDL at the crossroads between the maintenance of metabolic homeostasis and the immune-inflammatory response [106,107].

### 3.2. HDL, Lipid Rafts, and Immunomodulatory Function

Physiologically, a large percentage of cellular free cholesterol is localized in the plasma membrane, where it is distributed in a heterogeneous manner and enriched in the so-called lipid raft domains, which constitute ~30–40% of the mammalian cell membranes. These structures are highly dynamic, finely regulated, and essential for cellular functions, as they serve as a platform where proteins involved in signal transduction are transiently assembled or spatially organized to promote kinetically favorable interactions and generate an appropriate response to a stimulus. Indeed, many enzymes and proteins involved in the signal transduction system are active exclusively when clustered within lipid rafts. In the context of immune cell function, rafts organize signaling by many receptors and accessory proteins, including toll like receptors (TLRs), T and B cell receptors (TCR and BCR), and major histocompatibility (MHC) class II [86].

Cellular responses regulated by lipid rafts range from physiological to pathological; when cholesterol influx is high and exceeds the efflux capacity, the accumulation of cholesterol in membranes is believed to lead to cell activation and inflammation [108]. The abundance and functional properties of lipid rafts change very quickly in response to a stimulus, and thus cholesterol homeostasis is essential to ensure an optimal composition of lipid rafts and preserve the normal cell functions. Alterations in cholesterol trafficking at plasma membrane may modulate immune cell functions, including the recruitment and proliferation of monocytes, the expression of inflammatory cytokines in macrophages, and the activation and proliferation of lymphocytes [108,109].

### 3.3. HDL, Lipid Rafts in Innate Immune Cells

Macrophages derive from circulating monocytes, which are in turn generated following differentiation from hematopoietic stem and multipotential progenitor cells (HSPCs) in the bone marrow and other medullary organs [110]. Cholesterol metabolism plays a central role in hematopoietic precursors; in fact, these monocytes may bear a pre-programed ability to become M1-like macrophages (pro-inflammatory cells) once they enter the atherosclerotic lesion. On the other hand, removal of cholesterol from cells via HDL can regulate the production of innate immune cells, by acting on HSPCs proliferation [111]. Moreover, clinical conditions associated with very low levels of HDL-C reflect into changes in the balance of circulating monocytes subsets [112].

Cholesterol trafficking at plasma membrane and innate immunity signaling are closely linked; in fact, the cholesterol content of membrane rafts regulates cellular signaling by determining the selective localization of TLRs, involved in macrophage activation. TLR4 senses LPS and clusters within lipid rafts with CD14 to promote monocyte activation (Figure 1A) [113]. ABC transporter-deficient macrophages present with a free cholesterol accumulation in membranes that associates with an enhanced expression of inflammatory and chemokine genes in response to LPS, due to an increased cell surface expression of TLR4; this response is dampened following cholesterol depletion [114]. 

HDL exert an anti-inflammatory effect in macrophages even beyond their effect on cellular cholesterol levels. Indeed, HDL induce the expression and activation of activating transcription factor 3 (ATF3), a transcriptional modulator that provides negative feedback on TLR signaling (Figure 1A) [115]. In murine bone marrow-derived macrophages (BMDM), human peripheral blood mononuclear cells (PBMC), and in an in vivo model of acute inflammation (TLR ligand-treated mice), pretreatment with native HDL or reconstituted HDL (apoA-I and phospholipids) inhibited TLR-induced production of pro-inflammatory cytokines from macrophages [115]. This inhibitory effect was not due to the interaction of HDL with TLR ligands (contrarily to what observed for LPS, which is sequestered by HDL), and did not involve neither the modulation of TLR signaling events, nor the disruption of cytokine secretory processes, but instead was dependent on HDL-mediated induction of ATF3, which in turn suppressed the ability of TLRs to induce the expression of pro-inflammatory cytokines at a transcriptional level [115]. 

The modulation of lipid raft abundance and composition by HDL impacts cholesterol homeostasis also in HSPCs, and thus regulates the commitment to monocytes/macrophages and influences their function. Disruption of cholesterol homeostasis or prolonged exposure to hypercholesterolaemia increases the commitment of HSPCs to monocytes, resulting in monocytosis [111,116]. Similarly, loss of ABCA1, ABCG1, or apoE expression has been shown to foster hematopoietic cell proliferation by increasing cholesterol within lipid rafts, thus enhancing immune cell responsiveness. Similarly, the deficiency in the bone marrow of ABCG4 (another transporter involved in cholesterol efflux) increased cell surface expression of thrombopoietin (TPO) receptor (c-MPL) and enhanced megakaryocyte proliferation [117]; although these pathways have been connected with an increased risk of atherosclerosis, due to monocytosis, neutrophilia, and thrombocytosis, they also contribute to autoimmune diseases. This is supported by the evidence that the expansion of bone marrow cells observed in experimental models of RA correlates with a decreased expression of apoE, ABCA1, and ABCG1 in hematopoietic cells, that couples with an increased expression of key myeloid promoting growth factor receptors; this suggests that the systemic inflammation associated with the disease may, in turn, cause defective cellular cholesterol metabolism [118].

Although many potentially beneficial effects of HDL on inflammation have been reported, some studies suggested also a role in supporting the activation of the immune response: as an example, HDL-mediated passive cholesterol depletion (i.e., independent on cholesterol transporters) and lipid raft disruption result in the activation of multiple protein kinase C isoforms, which favor the production of pro-inflammatory cytokines (TNF-α and IL-12) and the reduction of IL-10 [119]. In this context, also apoA-I was shown to utilize MyD88, a TLR adaptor, to control reverse cholesterol transport [120] and induce pro-inflammatory cytokine expression (MCP-1, TNF-α, IL-6, IL-1β) [120]. This finding was confirmed in vivo following the injection of apoA-I, which elicited the MyD88- and TLR4-dependent neutrophil recruitment at the injection site [120]. Thus, HDL/apoA-I may exert anti- or pro-inflammatory effects, depending on the adaptations occurring in cell cholesterol content; in LPS-stimulated macrophages, the anti-inflammatory effect of HDL is observed in the early phases and reflects reduced TLR4 levels and signaling, whereas they become pro-inflammatory molecules under extensive cholesterol depletion conditions, reflecting a reduced interferon receptor signaling [121]. 

### 3.4. HDL, Cholesterol, and Lymphocyte Activation and Proliferation

Lymphocyte activation results from the presentation of an antigen bound to MHC-II molecules by antigen-presenting cells (APCs) [122]. APCs include macrophages and dendritic cells (DCs). The impairment of cholesterol efflux in DCs lacking ABCA1/G1 leads to cholesterol accumulation, inflammasome activation, and increased production of additional cytokines which result in enhanced T cell activation [123]. HDL and apoA-I can modulate the ability of APCs to activate T cells by cholesterol depletion from lipid rafts [124]. Furthermore, HDL suppress the activation, maturation, and cytokine secretion by DCs, which associate to a reduced expression of MHC-II and co-stimulatory molecules that are required for an efficient T cell activation [125,126]. In stimulated APCs, HDL downregulate the inflammatory response by interfering with the MyD88-dependent TLR4 signaling pathway, an effect that was mediated by ABCA1 and SR-BI, whereas ABCG1 did not appear to play a major role [126]. Similar to macrophages, also in DCs ATF3 expression was increased after exposure to HDL [126]. In addition, by inhibiting the oxidation of LDL, HDL may play an indirect role in regulating DC function; indeed, oxidized LDL were shown to promote the development of mature DCs [127,128]. Accordingly, changes in HDL function/levels typically reported in autoimmune disease (Table 1) may, at least in part, contribute to the alteration of DC functions observed in these pathological conditions [129].

Not only MHC-II, but also B-cell receptor (BCR) and T-cell receptor (TCR) are localized in lipids rafts rich areas. In T cells, cholesterol plasma membrane mediates TCR clustering, inhibits spontaneous TCR activation, and reduces TCR mobility when the conditions for activation are not appropriate (Figure 1B). Interestingly, distinct plasma membrane lipid profiles are associated with specific T helper subsets (Th1, Th2, or Th17), suggesting that lipid composition of cell membrane contributes to define functional T cell phenotypes [130].

Cholesterol is known to stabilize raft membrane domains and binds to the TCRβ-chain to facilitate TCR dimerization, thus increasing its avidity towards antigens. Hypercholesterolaemia enhances TCR stimulation and facilitates proliferative T cell response. These changes are associated also with alterations in regulatory T (Treg) cell development [131]. In resting T-cells, TCR and CD3 are localized in non-raft domains; following cholesterol enrichment within lipid rafts, TCR and related signaling proteins converge in these domains, an event which facilitates the formation of immunological synapses and the activation of downstream pathways [132]. Lipid raft integrity is thus crucial for T cell activation; the hyperactivity and hypersensitivity of T cells observed in autoimmune diseases, such as systemic lupus erythematosus, are associated with membrane cholesterol enrichment in lipid rafts and facilitate TCR-dependent signaling [133]. 

In an experimental model, the administration of exogenous apoA-I or recombinant HDL attenuates arthritis in an ABCA1-dependent manner [134]. This activity appears to be associated with a HDL-mediated modulation of maturation and function of dendritic cells, which suppresses T cell proliferation in vitro [126]. In an experimental model of dyslipidaemia-induced dermatitis, the ectopic macrophage apoA-I expression reduced cholesterol accumulation and CD4+ T cell levels in aortic lesions, skin, and skin-draining lymph nodes (LNs) without affecting HDL-C levels or tissue macrophage levels [135].

Tregs are a subset of CD4 T cells involved in immune homeostasis through the inhibition of the immune response in several cell types, including macrophages, APCs, and T cells, and have been shown to be atheroprotective. Still, in the context of atherosclerosis, Treg levels increase [136,137]. Hypercholesterolaemia, indeed, promotes TCR stimulation in CD4^+^ T cells and facilitates proliferative T cell response, and these effects are associated with a higher Treg population development in the thymus of mice fed with cholesterol-containing diet [131]. 

In mice with T cell-specific deficiency of ABCG1, an increased differentiation of naïve CD4 T cells into Tregs has been reported, protecting mice against atherosclerosis [138]; on the other hand, in *Ldlr^−/−^ApoAI*^−/−^ mice, which exhibit an autoimmune phenotype (with large spleens and peripheral lymph nodes enriched in cholesterol, expanded populations of T cells, B cells, DCs, and macrophages, and increase in T cell proliferation and activation), apoA-I infusion decreased the numbers of immune cells in LNs, increased the number of Tregs, and reduced lipid accumulation in LNs and skin [139]. Accordingly, apoA-I injection in *apoE*^−/−^ mice reduces intracellular cholesterol levels in Tregs and prevents their conversion into pro-atherogenic T follicular helper cells [140]. Tregs have also an elevated capacity to bind and internalize HDL, likely due to the high expression of SR-BI compared with other CD4 cell subsets [141]. The internalization of HDL increases the survival of Tregs through several mechanisms [141]. 

Altogether, these observations suggest that an optimal level of plasma membrane cholesterol is crucial for Treg development and/or function, and that variations in cholesterol levels may alter Treg homeostasis.

## 4. Genetics as a Proxy to Study the Impact of HDL Cholesterol and Functions in Inflammation and Immune Disorders

A powerful tool to appreciate the role of HDL comes from studies on monogenic disorders resulting in extremely low or high HDL-C levels. Monogenic disorders associated with low HDL-C levels include those related to loss-of-function mutations in *APOA1*, the gene encoding apoA-I, *LCAT*, encoding lecithin-cholesterol acyl transferase (LCAT), and *ABCA1*, encoding ABCA1. Genes implicated in monogenic disorders associated with high HDL-C levels include *CETP*, encoding cholesteryl ester transfer protein (CETP), *SCARB1*, encoding scavenger receptor class B member 1 (SRB1), and *LIPC*, encoding hepatic triacylglycerol lipase [142]. Patients with Tangier disease, despite having very low HDL-C levels due to ABCA1 deficiency, do not manifest premature coronary artery disease, but rather present an increased inflammatory status [143]; this has been associated to an increased inflammasome activation triggered by reduced cholesterol efflux from myeloid cells [144]. Moreover, mendelian randomization studies failed to demonstrate a causal link between HDL-C and CVD risk [145], but rather indicate that low HDL-C levels increase the risk of infections [6]. In addition, elevated peripheral blood leukocyte counts were reported in subjects with low HDL-C levels of any origin [146]. In line with this observation, subjects carrying a low HDL-C polygenic score have been shown to present an increased risk of hospitalization for infections [147]; this is consistent with the increased mortality for sepsis reported in subjects with gain-of-function mutation on *CETP*, who present a dramatic reduction of HDL-C levels during infection [148]. It is tempting to speculate that CETP, beyond transferring cholesteryl esters from HDL to apolipoprotein (apo) B-containing triglyceride-rich lipoproteins (TRLs), could also participate to immune response following a bacterial insult. This hypothesis is supported by the evidence that CETP is predominantly produced by Kupffer cells, specialized liver macrophages, and that its expression is reduced during inflammation; in turn, this leads to an increase in HDL-C levels that may contribute to LPS clearance during infections [149]. Similarly, LCAT deficiency in mice was associated with the presence of immature discoidal HDL and a reduced LPS-neutralizing capacity [150], thus suggesting that not only the concentration but also the type of HDL subclasses differently modulates immune responses [151].

## 5. Conclusions

HDL are complex particles constituted by different proteins and lipids, also acting as cargo of other molecules, such as hormones and vitamins. The aspects highlighting a role for HDL in immune-inflammatory diseases support the investigation of HDL-based therapies in the context of these diseases. However, clinical trials and genetic studies have demonstrated that raising HDL-C levels would not have beneficial effects, thus directing pharmacology toward the identification of therapeutic approaches able to mirror HDL protective functions. Recombinant HDL and HDL mimetics have shown favorable effects in increasing cholesterol efflux [152,153], improving endothelial functions [154], and reducing inflammation [155], that might prove to be beneficial beyond atherosclerosis [156,157,158]. Parallel to increasing small, discoidal HDL, which possess the highest cholesterol efflux potential [159], interfering with HDL maturation might be beneficial in the context of inflammatory diseases. This is the case of CETP inhibitors that, despite the failure in preventing cardiovascular events, have been shown to increase the survival rate in humanized models of sepsis [160], or recombinant human LCAT, that, by reducing HDL-bound SAA and increasing HDL-bound apoA-I as well as HDL functionality [161], could exert protective effect also against SARS-Cov2 infections [162]. Indeed, increased COVID-19 disease severity and worse prognosis was reported in patients with low HDL-C levels [163,164,165]; in addition, major changes in HDL proteome and decreased functionality have been described in severe COVID-19 patients [166].

Therefore, the identification of the molecular mechanisms that lead to HDL dysfunction in immune-inflammatory diseases would offer the opportunity to design intervention approaches aimed at restoring HDL functionality, representing a complementary approach to current anti-inflammatory treatments. 

## Figures and Tables

**Figure 1 cells-10-01061-f001:**
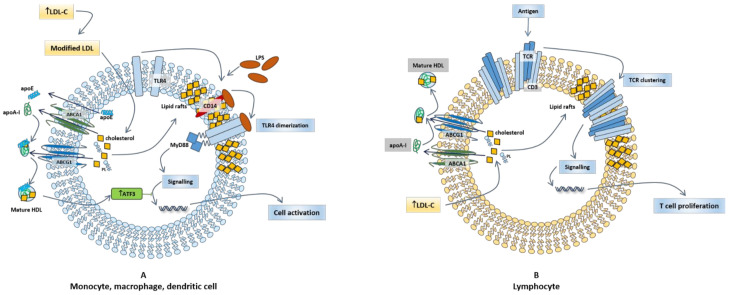
Role of cholesterol efflux in innate immune cell activation. (**A**) TLR4, which is expressed in innate immune cells including monocytes, macrophages and dendritic cells, is activated by lipopolysaccharide (LPS). These cells express also high amounts of CD14, which facilitates the activation of TLR4 by LPS. LPS induces a transient TLR4 trafficking to lipid rafts; following dimerization, a process required for the initiation of signaling during innate immune response, TLR4 triggers the MyD88-dependent signaling pathway, leading to the production of pro-inflammatory cytokines and cell activation. ApoA-I and HDL, as well as apoE, may dampen inflammation by selectively reducing the free cholesterol content in lipid rafts and the consequent chance of an MyD88-dependent TLR trafficking to lipid rafts in cells exposed to LPS. PL: phospholipids. (**B**) In resting T-cells, TCR is localized in non-raft domains of plasma membrane in monomeric form. Cholesterol is imported into the cell LDL; when TCRs become antigen stimulated, they associate with lipid rafts, thus resulting in the activation of downstream signaling pathways and consequent T cell proliferation. ApoA-I and HDL, by removing cholesterol from cell membrane, may modulate the abundance of lipid rafts and their content of cholesterol, thus controlling T cell immune response.

**Table 1 cells-10-01061-t001:** Major HDL alterations in selected autoimmune diseases.

Autoimmune Disease	HDL Alterations (Level/Function)	Reference
Systemic lupus erythematosus	Reduced levels of HDL-C	de Carvalho JF et al., 2008 [31]; Hahn BH et al., 2008 [32]; Toloza SMA et al., 2004 [33]; Gamal SM et al., 2017 [34]
	No difference in HDL-C levels	
	“Lupus dyslipoproteinemia”	Yuan J et al., 2016 [35]
	Reduced PON1 activity	Kiss E et al., 2007 [36]; Gaal K et al., 2011 [37]
	Pro-inflammatory HDL (piHDL) with increased SAA content, decreased apoA-I levels	McMahon M et al., 2011 [38]; G HB et al., 2011 [39]; Van Lenten BJ et al., 2001 [40]
	Oxidized HDL	Smith CK et al., 2014 [41]
	Reduced HDL particles/size distribution	Chung CP et al., 2008 [42]; Hua X et al., 2009 [43]; Juarez-Rojas J et al., 2008 [44]
	Reduced CEC	Smith CK et al., 2014 [41]; Skaggs BJ et al., 2010 [45]
	Reduced anti-inflammatory potential	Smith CK et al., 2016 [46]
Rheumatoid arthritis	Reduced levels of HDL-C	Jae-Yong Kim et al., 2016 [47]; Myasoedova E et al., 2011 [48]
	No difference in HDL-C	Chung CP et al., 2005 [49]; Liao KP et al., 2013 [50]
	Pro-inflammatory HDL (piHDL)	Hahn BH et al., 2008 [32]; Charles-Schoeman C et al., 2009 [51]; Watanabe J et al., 2012 [52]
	Oxidized HDL	Vivekanandan-Giri A et al., 2013 [53]
	Reduced anti-oxidative activity	Gomez Rosso L et al., 2014 (in patients with active disease) [54]
	Reduced CEC	Beatriz Tejera-Segura et al., 2017 (in patients with low or moderate disease activity compared to patients in remission) [55]; Charles-Schoeman C et al., 2012 [56]; Ronda N et al., 2014 [57]
Psoriasis	Increased HDL-C levels	Tam LS et al., 2008 [58]; Mallbris L et al., 2006 [59]; Borska L et al., 2017 [60]; Nakhawa YC et al., 2014 [61]
	Decreased HDL-C levels	Pietrzak A et al., 2019 [62]; Pietrzak A et al., 2017 [63]; Akkara Veetil BM et al., 2012 [64]
	Unchanged HDL-C levels	Uyanik BS et al., 2002 [65]; Asha K et al., 2017 [66]; Miller IM et al., 2013 [67]
	Change in HDL particle size	Yu Y et al., 2012 [68]; Tom WL et al., 2016 [69]; Wolk R et al., 2017 [70]
	Pro-inflammatory HDL (piHDL)	He L et al., 2014 [71]; Staniak HL et al., 2014 [72]
	Reduced CEC	Holzer M et al., 2012 [24]; Tom WL et al., 2016 [69]; Holzer M et al., 2014 [73]; Mehta NN et al., 2012 [74]
Chron’s disease	Decreased HDL-C levels and biochemical changes in HDL particles	Romanato G et al., 2009 [10]
Allergic asthma	Lower HDL levels (only children)	Peng J et al., 2017 [75]
	Inverse correlation between HDL-C levels and circulating eosinophils and monocytes	Barochia AV et al., 2017 [76]; Rastogi D et al., 2015 [77]
	Positive correlation of serum apoA-I levels with less severe airflow obstruction in asthmatic individuals	Cirillo DJ et al., 2002 [78]
	ApoA-I levels decreased in bronchoalveolar lavage fluid of patients with moderate to severe asthma	Park SW et al., 2013 [79]
Atopic dermatitis	Increased HDL-C levels in patients	Schäfer T et al., 2003 [80]
	No difference in a paediatric population	Agón-Banzo PJ et al., 2020 [81]
	HDL enrichment in apoA-II, SAA, and phosphatidylinositol and significant reduction in the content of apoC-III, apoE, cholesteryl ester, free cholesterol, lysophosphatidylcholine, and phosphatidylethanolamine	Trieb et al., 2019 [82]

## Data Availability

Not applicable.

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
