# Peer review of "HDL in Immune-Inflammatory Responses: Implications beyond Cardiovascular Diseases"

_cells, 2021, doi:10.3390/cells10051061_

Round 1

Reviewer 1 Report

Overall:

Bonacina et al. have addressed an important issue on the role of HDL in inflammatory responses in cardiovascular and other diseases. The manuscript is very well written with very minimum linguistic errors. The authored have discussed interesting issues on the role of HDL in severity of several diseases in a cellular level and overall, the manuscript is timing and is beneficial to the field.

I have only following minor comments and suggestions:

The title of the manuscript refers to inflammatory responses and the readers would expect to have at least a brief information regarding all aspect of inflammation associated with HDL. Recent findings demonstrate that beside HDL‐cholesterol levels, HDL function also contributes in cardiovascular risk and possibly other diseases severity such as sepsis.

-I would suggest adding a section explaining HDL modification such as dysfunctional HDL on inflammatory responses. HDL Modifications by SDMA or reactive aldehydes such as acrolein (Acro), 4‐hydroxynonenal, and malondialdehyde (MDA) have been shown to dysfunction HDL.

-As the authors are aware, dysfunctional HDL also could contribute cardiovascular disease and particularly chronic kidney diseases.  Would the authors include this type in the text?

-Based on the strong experience of the authors, I would suggest adding a table, which summarizes the role of modified HDL in diseases severity?  

The level of HDL have been associated to atopic allergic diseases and atopic dermatitis. Could the authors include the data in the table 1.

Many scientists have discussed the protective effect of HDL level in infection diseases and it could be included in the manuscript too. I appreciate the data about protective effect of HDL on COVID-19 severity.  However, COVID-19 has not been addressed in anyother part of the text while almost half of the conclusions section devoted to COVID-19. I would strongly suggest to make a separate section about the role of HDL on infectious diseases severity and shorten the COVID-19 part in conclusions section.    

Author Response

Reviewer 1

Dear Reviewer 1,

in this revised version of the manuscript we have addressed your comments and those raised by Reviewer 2.

Concerning the specific aspects you raised, please find below our replies:

The title of the manuscript refers to inflammatory responses and the readers would expect to have at least a brief information regarding all aspect of inflammation associated with HDL. Recent findings demonstrate that beside HDLcholesterol levels, HDL function also contributes in cardiovascular risk and possibly other diseases severity such as sepsis

Your raised a very important point about the contribution of HDL in immune-inflammatory response. This, however, will be the topic of another review for the same special issue and therefore we have decided not to discuss this aspect in our work but rather refer to the other review in the same issue.

I would suggest adding a section explaining HDL modification such as dysfunctional HDL on inflammatory responses. HDL Modifications by SDMA or reactive aldehydes such as acrolein (Acro), 4hydroxynonenal, and malondialdehyde (MDA) have been shown to dysfunction HDL

Your suggestion is well taken; the relation between dysfunctional HDL and inflammatory responses is now mentioned in section 2.1.

As the authors are aware, dysfunctional HDL also could contribute cardiovascular disease and particularly chronic kidney diseases. Would the authors include this type in the text?

Thank you for raising this point, we have mentioned this aspect in the revised version of the manuscript (see paragraph 2.1). Please note that in the same Special Issue, a contribution dedicated on the topic “HDL and the kidney” has been just released (High-Density Lipoproteins and the Kidney. Strazzella A, Ossoli A, Calabresi L. Cells. 2021 Mar 31;10(4):764. Doi: 10.3390/cells10040764.)doi.org/10.3390/cells10040764

-Based on the strong experience of the authors, I would suggest adding a table, which summarizes the role of modified HDL in diseases severity? The level of HDL have been associated to atopic allergic diseases and atopic dermatitis. Could the authors include the data in the table 1.

Thank you for this suggestion. In table 1, we have now included info on HDL-C levels and HDL functionality in patients with different autoimmune diseases. Moreover, we have also commented on HDL levels/function in allergic diseases and atopic dermatitis.

Many scientists have discussed the protective effect of HDL level in infection diseases and it could be included in the manuscript too. I appreciate the data about protective effect of HDL on COVID-19 severity. However, COVID-19 has not been addressed in any other part of the text while almost half of the conclusions section devoted to COVID-19. I would strongly suggest to make a separate section about the role of HDL on infectious diseases severity and shorten the COVID-19 part in conclusions section.

We agree with the Reviewer that the role of HDL in infection has been only mentioned but not in deeply commented. As said before, the treatment of this topic is out of the scope of the present review as another manuscript in this special issue has focused on the role of HDL during infection. For consistency we have indeed shortened the section of COVID in the conclusion.

Reviewer 2 Report

In this manuscript, Bonacina et al reviewed recent advances on the properties of HDL on the modulation of immune/inflammatory responses. The paper is clearly written, very interesting and timely.

There following are some concerns:

  • Line 61 CETP. Please give the detail of the abbreviation
  • As discussed by the authors in the section 3.4, HDL strongly inhibit the proTh1 function of mature DC, characterized by their ability to induce IFNg secretion by T cells. This observation was initially published in 2012:
  1. Perrin-Cocon, O. Diaz, M. Carreras, S. Dollet, A. Guironnet-Paquet, P. Andre, V. Lotteau, High-density lipoprotein phospholipids interfere with dendritic cell Th1 functional maturation, Immunobiology 217 (2012) 91e99.

I think that this key reference should be added.

  • Moreover, by controling LDL oxidation, HDL play an indirect role in the regulation of DC function, since oxLDL promote DC maturation and a polarized Th1 response (PMID: 11564795, 14688309). This point should be discussed, and could be connected to the altered DCs functions observed in autoimmune diseases (PMID: 27652503).

Author Response

Reviewer 2

In this revised version of the manuscript we have addressed your comments and those raised by Reviewer 1.

Concerning the specific aspects, you raised, please find below our replies:

Line 61 CETP. Please give the detail of the abbreviation

Your point is well taken and CETP is now spelled out.

As discussed by the authors in the section 3.4, HDL strongly inhibit the proTh1 function of mature DC, characterized by their ability to induce IFNg secretion by T cells. This observation was initially published in 2012:

  1. Perrin-Cocon, O. Diaz, M. Carreras, S. Dollet, A. Guironnet-Paquet, P. Andre, V. Lotteau, High-density lipoprotein phospholipids interfere with dendritic cell Th1 functional maturation, Immunobiology 217 (2012) 91e99.

I think that this key reference should be added.

Your suggestion is well taken and the suggested reference has been included in the revised version of the paper (see page 7, ref. 125)

Moreover, by controlling LDL oxidation, HDL play an indirect role in the regulation of DC function, since oxLDL promote DC maturation and a polarized Th1 response (PMID: 11564795, 14688309). This point should be discussed, and could be connected to the altered DCs functions observed in autoimmune diseases (PMID: 27652503).

Thank you for bringing this point, the topic of these papers has been briefly discussed (see page 8, refs. 127-129)